# Capecitabine in Combination with Endocrine Therapy as Maintenance Therapy after Bevacizumab Plus Paclitaxel Induction Therapy for Hormone Receptor-Positive, HER2-Negative Metastatic Breast Cancer: KBCSG-TR1214

**DOI:** 10.3390/cancers13174399

**Published:** 2021-08-31

**Authors:** Norikazu Masuda, Tetsuhiro Yoshinami, Masahiko Ikeda, Makiko Mizutani, Miki Yamaguchi, Yoshifumi Komoike, Tsutomu Takashima, Katsuhide Yoshidome, Junji Tsurutani, Mitsuhiko Iwamoto, Fumie Fujisawa, Hiroyuki Yasojima, Jun Yamamura, Hirotaka Morishima, Fuminori Aki, Tomomi Yamada, Satoshi Morita, Takahiro Nakayama

**Affiliations:** 1Department of Surgery, Breast Oncology, National Hospital Organization Osaka National Hospital, 2-1-14 Hoenzaka, Chuou-ku, Osaka-shi 540-0006, Osaka, Japan; mizutani.makiko.ay@mail.hosp.go.jp (M.M.); yasojima.hiroyuki.wt@mail.hosp.go.jp (H.Y.); 2Department of Medical Oncology, Osaka International Cancer Institute, 3-1-69 Otemae, Chuo-ku, Osaka-shi 541-8567, Osaka, Japan; yosinami-te@onsurg.med.osaka-u.ac.jp (T.Y.); fujisawa-fu@mc.pref.osaka.jp (F.F.); 3Breast and Endocrine Surgery, Graduate School of Medicine, Osaka University, 2-2-E10 Yamadaoka, Suita-shi 565-0871, Osaka, Japan; 4Department of Breast and Thyroid Surgery, Fukuyama City Hospital, 5-23-1 Zao-cho, Fukuyama-shi 721-8511, Hiroshima, Japan; masahikoikeda@city.fukuyama.hiroshima.jp; 5Department of Breast Surgery, JCHO Kurume General Hospital, 21 Kushiharamachi, Kurume-shi 830-0013, Fukuoka, Japan; yamaguchi-miki@kurume.jcho.go.jp; 6Department of Medicine, Kindai University Hospital, 377-2 Ohnohigashi, Osaka-Sayama-shi 589-8511, Osaka, Japan; komoike@med.kindai.ac.jp; 7Department of Breast and Endocrine Surgery, Graduate School of Medicine, Osaka City University, 1-4-3 Asahi-machi, Abeno-ku, Osaka-shi 545-8585, Osaka, Japan; m1158973@med.osaka-cu.ac.jp; 8Department of Breast and Endocrine Surgery, Osaka Police Hospital, 10-31 Kitayama-cho, Tennouji-ku, Osaka-shi 543-0035, Osaka, Japan; yoshidome-k@umin.ac.jp; 9Advanced Cancer Translation Research Institute, Showa University, 1-5-8 Hatanodai, Shinagawa-ku 142-8555, Tokyo, Japan; tsurutaj@med.showa-u.ac.jp; 10Breast and Endocrine Surgery, Osaka Medical College, 2-7 Daigakumachi, Takatsuki-shi 569-8686, Osaka, Japan; sur067@poh.osaka-med.ac.jp; 11Department of Surgery, Sakai City Medical Center, 1-1-1 Ebaraji-cho, Nishi-ku, Sakai-shi 593-8304, Osaka, Japan; june10th@zeus.eonet.ne.jp; 12Department of Breast Surgery, Osaka Rosai Hospital, 1179-3 Nagasone-cho, Kitaku, Sakai-shi 591-8025, Osaka, Japan; morisima@bd5.so-net.ne.jp; 13Ito Surgical Mammary Gland Clinic, 12-10 Fudaba, Kochi-shi 781-0085, Kochi, Japan; suito@basil.ocn.ne.jp; 14Department of Medical Innovation, Osaka University Hospital, 2-2, Yamadaoka, Suita-shi 565-0871, Osaka, Japan; tomomi.yamada@dmi.med.osaka-u.ac.jp; 15Department of Biomedical Statistics and Bioinformatics, Kyoto University Graduate School of Medicine, 54 Kawahara-cho, Shogoin, Sakyo-ku, Kyoto-shi 606-8507, Kyoto, Japan; smorita@kuhp.kyoto-u.ac.jp; 16Department of Breast and Endocrine Surgery, Osaka International Cancer Institute, 3-1-69 Otemae, Chuo-ku, Osaka-shi 541-8567, Osaka, Japan; taqnakayama@gmail.com

**Keywords:** advanced and metastatic breast cancer, bevacizumab, capecitabine, endocrine therapy, hormone receptor positive, HER2-negative breast cancer, maintenance therapy, randomized trial

## Abstract

**Simple Summary:**

To investigate a possible treatment strategy for hormone receptor (HR)-positive, HER2-negative advanced and/or metastatic breast cancer (AMBC), we investigated the clinical usefulness of adding capecitabine to maintenance endocrine therapy after induction chemotherapy and the efficacy of reinduction chemotherapy. Patients who had received bevacizumab–paclitaxel induction therapy and did not have progressive disease were randomized to receive maintenance therapy with endocrine therapy alone (group E; *n* = 46) or endocrine therapy plus capecitabine (group EC; *n* = 44). The median progression-free survival (PFS) under maintenance therapy (primary endpoint) was significantly longer in group EC than in group E (11.1 vs. 4.3 months; hazard ratio, 0.53; *p* < 0.01). At 24 months from the induction therapy start, the overall survival (OS) rate was significantly higher in group EC than in group E (83.5% vs. 62.3%; *p* = 0.02). Therefore, the addition of capecitabine to maintenance endocrine therapy may be a beneficial option after induction chemotherapy for HR-positive, HER2-negative AMBC patients.

**Abstract:**

Optimal treatment strategies for hormone receptor (HR)-positive, HER2-negative advanced and/or metastatic breast cancer (AMBC) remain uncertain. We investigated the clinical usefulness of adding capecitabine to maintenance endocrine therapy after induction chemotherapy and the efficacy of reinduction chemotherapy. Patients who had received bevacizumab–paclitaxel induction therapy and did not have progressive disease (PD) were randomized to maintenance therapy with endocrine therapy alone (group E) or endocrine plus capecitabine (1657 mg/m^2^/day on days 1–21, q4w) (group EC). In case of PD after maintenance therapy, patients received bevacizumab–paclitaxel reinduction therapy. Ninety patients were randomized. The median progression-free survival (PFS) under maintenance therapy (primary endpoint) was significantly longer in group EC (11.1 {95% CI, 8.0–11.8} months) than in group E (4.3 {3.6–6.0} months) (hazard ratio, 0.53; *p* < 0.01). At 24 months from the induction therapy start, the overall survival (OS) was significantly longer in group EC than in group E (hazard ratio, 0.41; *p* = 0.046). No difference was found in the time to failure of strategy (13.9 and 16.6 months in groups E and EC, respectively). Increased capecitabine-associated toxicities in group EC were tolerable. Addition of capecitabine to maintenance endocrine therapy may be a beneficial option after induction chemotherapy for HR-positive, HER2-negative AMBC patients.

## 1. Introduction

The aim of treatment for advanced and/or metastatic breast cancer (AMBC) is to prolong patients’ survival and improve their quality of life (QOL) by controlling disease symptoms. For hormone receptor (HR)-positive, human epidermal growth factor receptor 2 (HER2)-negative breast cancer, which is the most common subtype of advanced breast cancer, endocrine therapy, as many regimens as possible, is recommended [1]. Moreover, adding molecular targeted drugs, such as the recently developed cyclin-dependent kinase 4 and 6 (CDK4/6) inhibitors, to endocrine therapy has become recent standard first- or second-line treatment [2,3,4,5,6,7]. However, as resistance to these drugs may develop, or in case of life-threatening conditions, such as visceral involvement, chemotherapy is the treatment of choice. In the hope of a rapid response, bevacizumab in combination with an anticancer drug is considered.

Bevacizumab has been shown to significantly increase progression-free survival (PFS) and improve the response rate when combined with docetaxel [8,9] or paclitaxel [10,11] in AMBC patients. In Japan, weekly paclitaxel plus bevacizumab is widely used as one of the standard therapy regimens for HER2-negative metastatic breast cancer [12]. However, adverse events (AEs) of these drugs, such as peripheral neuropathy, hypertension, and proteinuria, are a concern as they may reduce patients’ QOL over the long term. In general, a longer duration of first-line chemotherapy is associated with prolonged PFS and overall survival (OS) [13], so intensive treatment tends to be continued until disease progression or intolerable toxicity in daily clinical practice, while AEs need to be minimized to maintain patients’ QOL.

Maintenance therapy may be one way to achieve this balance. At the Fifth International Consensus Conference for Advanced Breast Cancer, experts agreed that in cases of HR-positive, HER2-negative AMBC in which induction chemotherapy has been effective, endocrine maintenance therapy is a reasonable treatment option [14,15]. Induction therapy followed by maintenance therapy is expected to improve survival and QOL in AMBC.

Furthermore, capecitabine is expected to be beneficial for the treatment of AMBC. The IMELDA [16] and CREATE-X [17] trials have shown the efficacy of capecitabine against high-risk breast cancer in cases of non-cross-resistance with intravenous chemotherapeutic agents, with good compliance. For HR-positive, HER2-negative AMBC, the usefulness of the addition of capecitabine to bevacizumab as maintenance therapy has been suggested [16,18]; however, the current evidence regarding effective maintenance therapy is limited and needs to be explored.

Additionally, the strategy after maintenance therapy is also important in view of how effectively anticancer drugs can be used in the treatment of AMBC. A Japanese study has shown that reinduction therapy with bevacizumab and paclitaxel, following temporal discontinuation of paclitaxel due to its AEs, is safe and effective [19].

Based on these findings, we conducted the present multicenter randomized phase 2 trial (KBCSG-TR1214) with the following aims: (1) to investigate the clinical usefulness of the addition of capecitabine to endocrine therapy, compared with endocrine therapy alone, as maintenance therapy after induction chemotherapy with bevacizumab and paclitaxel and (2) to evaluate the efficacy of bevacizumab–paclitaxel reinduction therapy after failure of maintenance therapy in patients with HR-positive, HER2-negative AMBC.

## 2. Results

### 2.1. Patient Characteristics

A total of 116 patients were enrolled between August 2012 and April 2016 and received induction therapy with bevacizumab and paclitaxel. Figure 1 shows the patient flow. Of the 116 patients enrolled, 26 did not proceed to randomization, mainly due to progressive disease (PD). The remaining 90 patients were randomized to endocrine-only therapy (group E, *n* = 46) or endocrine therapy plus capecitabine (group EC, *n* = 44). Allocation to treatment groups ended in October 2016.

Patients in the two groups were well balanced in terms of baseline characteristics (Table 1). The median age was 59.8 years (range, 41.5–75.8 years) in group E and 59.9 years (range, 34.5–81.0 years) in group EC. In each group, just over a fifth of the patients were premenopausal. Of the 90 patients randomized, 85 (94.4%) had a measurable lesion, and 81 (90.0%) had received no prior chemotherapy for AMBC and therefore received bevacizumab plus paclitaxel as first-line chemotherapy. Most (75/90, 83.3%) had completed 6 cycles of bevacizumab–paclitaxel induction therapy, and the remaining 15 (16.7%) had received 4 or 5 cycles (data not shown).

Most patients in each group received endocrine therapy with an aromatase inhibitor, or if premenopausal, an aromatase inhibitor plus a luteinizing-hormone-releasing hormone (LHRH) agonist: 28 (60.9%) of the 46 patients in group E and 27 (61.4%) of the 44 patients in group EC (Table 2). A total of 81 of the 116 patients had complete response (CR) or partial response (PR), giving an objective response rate (ORR) of 69.8% (95% CI, 60.6–78.0%). The ORRs (combining CR and PR) to the bevacizumab–paclitaxel induction therapy were 76.1% (35/46) and 77.3% (34/44) in groups E and EC, respectively (Table 3).

At the data cutoff point (September 2018), the median follow-up period was 30.8 months (range, 6.1–72.1 months). By that time, 80 (88.9%) of the 90 patients had discontinued maintenance therapy (43/46, 93.5%, in group E, and 37/44, 84.1%, in group EC), all due to PD, except for 1 patient in group EC, which was due to AE. A total of 54 (67.5%) of the patients who discontinued maintenance therapy had been switched to bevacizumab–paclitaxel reinduction therapy, and 26 (13 from each group) had been switched to another therapy or breast-conserving surgery (Figure 1).

### 2.2. Efficacy

#### 2.2.1. PFS of Maintenance Therapy

The median PFS of maintenance therapy (primary endpoint) was significantly longer in group EC than in group E (11.1 months, 95% CI 8.0–11.8 months, versus 4.3 months, 95% CI 3.6–6.0 months; *p* < 0.01) (Figure 2). The hazard ratio (HR) for PFS was 0.53 (95% CI, 0.34–0.83). There was no significant difference in the ORR between group E (32.6%) and group EC (50.0%). The disease control rate (DCR) was higher in group EC than in group E (72.7%, 32/44 patients versus 50.0%, 23/46 patients; *p* = 0.03).

As for the ad hoc subgroup analysis, the factors affecting the PFS of maintenance therapy were investigated (Appendix A). The addition of capecitabine tended to be beneficial in all subgroups, particularly in patients who had previously received ≥3 regimens of endocrine therapy for AMBC and those in whom induction therapy had produced CR or PR.

#### 2.2.2. Efficacy of Reinduction Therapy

A total of 54 (67.5%) of the 80 patients who had discontinued maintenance therapy were switched to bevacizumab–paclitaxel reinduction therapy: 30 (65.2%) of the 46 patients in group E and 24 (54.5%) of the 44 patients in group EC. The median PFS, ORR, and DCR of bevacizumab–paclitaxel reinduction therapy after either of the two maintenance therapies were 7.8 months (95% CI, 6.7–9.5 months), 20.4% (11/54 patients), and 63.0% (34/54 patients), respectively. Nine patients (16.7%) did not respond to the reinduction therapy and were found to have PD.

The median PFS of bevacizumab–paclitaxel reinduction therapy tended to be longer in group E than in group EC; however, no significant difference was found between the groups (9.1 months, 95% CI 6.7–11.3 months, and 7.8 months, 95% CI 5.5–9.5 months, respectively; log-rank *p* = 0.053) (Figure 3).

#### 2.2.3. Time to Failure of Strategy

Analysis of data from the 90 patients randomized to maintenance therapy found no significant difference between the two groups in terms of the median time to failure of strategy (13.9 months, 95% CI 8.7–17.8 months, and 16.6 months, 95% CI 11.5–19.6 months, in groups E and EC, respectively) (Figure 4).

#### 2.2.4. OS from the Start of Induction Therapy

The median OS from the start of bevacizumab–paclitaxel induction therapy was significantly longer in group EC than in group E: 43.8 months (95% CI, 33.7 months to not estimable, NE, versus 34.9 months (95% CI 23.3 months to NE)) (HR: 0.41, 95% CI 0.17–0.99; log-rank *p* = 0.046). In groups EC and E, the OS rates were 83.5% (68.5–91.8%) and 62.3% (46.6–74.6%) (log-rank *p* = 0.02), respectively, at 24 months, and 62.6% (44.5–76.3%) and 47.6% (32.2–61.5%), respectively, at 36 months (Figure 5).

As for the ad hoc subgroup analysis, the factors affecting OS from the start of induction therapy were investigated (Appendix A). The addition of capecitabine to endocrine maintenance therapy benefitted all subgroups; in particular, a significant superiority to endocrine therapy alone was noted in patients who had previously received ≥3 regimens of endocrine therapy for AMBC.

### 2.3. Safety

All 90 patients assigned to maintenance therapy received at least one dose of the study treatment. Therefore, their data were included in the safety analysis set. In groups E and EC, 174 AEs (grade ≥3 AEs: 12) and 299 AEs (grade ≥3 AEs: 17) were recorded, respectively. Table 4 lists AEs that were experienced by ≥10% of the patients or grade ≥3. All cases of grade ≥3 AEs were manageable, and no patients discontinued the study treatment due to AEs.

In each group, sensory neuropathy and alopecia were experienced by most patients (>65%), and hypertension by more than a third. The incidence of AEs considered by the investigators to be related to capecitabine, namely, palmar–plantar erythrodysesthesia syndrome, fatigue, malaise, anorexia, nail disorders, skin changes, and dysgeusia, was higher in group EC than in group E (affecting ≥27.3% versus ≤15.2%). However, these AEs were grade ≥3 in only 7 cases (6 cases of palmar–plantar erythrodysesthesia syndrome and 1 case of fatigue) in group EC.

Regarding hematologic AEs, the incidence of anemia was similar in both groups: 17.4% (8/46) and 20.5% (9/44) in groups E and EC, respectively. Neutropenia was experienced in 22.7% in group EC but was grade ≥3 in 1 case.

Regarding serious AEs other than symptoms associated with worsening of disease, there was 1 case of bilateral pulmonary artery thrombus formation during maintenance therapy (outcome: improved) and 1 case of grade 4 intraventricular hemorrhage during bevacizumab–paclitaxel reinduction therapy.

## 3. Discussion

### 3.1. Maintenance Therapy: Clinical Usefulness of the Addition of Capecitabine to Endocrine Therapy

In the present study, we evaluated two maintenance therapies in patients with HR-positive, HER2-negative AMBC to determine the clinical usefulness of the addition of capecitabine to maintenance endocrine therapy, as compared with endocrine therapy alone, in patients who had received bevacizumab–paclitaxel induction therapy. The results showed that the addition of capecitabine extended the median PFS from 4.3 months to 11.1 months, representing a 47% reduction in risk of disease progression or death. In terms of tolerability and safety, the results showed both maintenance therapies to have manageable and tolerable toxicity. These findings are consistent with the previous literature suggesting that maintenance therapy with capecitabine may be beneficial in terms of PFS with preservation of QOL [20].

Previously, the IMELDA trial evaluated the efficacy of bevacizumab–capecitabine maintenance therapy, after bevacizumab–docetaxel induction therapy, in patients with HER2-negative metastatic breast cancer [16]; the median PFS was significantly longer in patients who received the bevacizumab–capecitabine regimen than in those who received bevacizumab only (11.9 months vs. 4.3 months; HR, 0.38), and an even more marked improvement was shown for the median OS (39.0 months vs. 23.7 months; HR: 0.43). In this trial, the docetaxel used for induction therapy was switched to capecitabine for maintenance therapy to avoid the AEs (including neuropathy and edema) that limit the long-term use of taxanes. Additionally, the AROBASE study [21] compared the effects of maintenance bevacizumab–exemestane therapy with those of continuous bevacizumab–taxane induction therapy; the median PFS from randomization was 7.6 (95% CI, 5.4–10.9) months in patients who were switched to the maintenance bevacizumab–exemestane regimen, and 8.1 (95% CI, 6.5–10.7) months in patients who continuously received the bevacizumab–taxane regimen. Although the results failed to show the superiority of maintenance therapy, subgroup analyses suggested that patients without a previous history of hormone resistance, especially in the case of treatment naïve to aromatase inhibitor therapy, may benefit from the maintenance strategy. Moreover, the Dutch Breast Cancer Research Group compared intermittent versus continuous first-line treatment with bevacizumab plus paclitaxel; unfortunately, this strategy was not successful to demonstrate the noninferiority of the intermittent regimen to continuous treatment in terms of PFS or OS [22]. Promising results have also been reported in a Japanese phase 2 trial in which patients underwent induction therapy with bevacizumab plus paclitaxel followed by maintenance therapy with eribulin (median PFS: 10.7 months; OS: 20.0 months) [23]. Therefore, previous studies examining maintenance therapy have not yielded consistent results, and thus there is a need to explore a maintenance therapy with an optimal balance of toxicity and efficacy.

The regimen used in the present study, that is, maintenance endocrine therapy with or without the addition of capecitabine, is unique in that bevacizumab was suspended during maintenance therapy. While continuous bevacizumab may be useful to prolong PFS, its long-term use is associated with proteinuria, which often prevents treatment continuation. To resolve this problem, we investigated a treatment strategy that includes bevacizumab–paclitaxel reinduction therapy so that bevacizumab can be suspended during maintenance therapy. The addition of capecitabine to endocrine maintenance therapy resulted in favorable PFS (11.1 months; 95% CI, 8.0–11.8 months). We consider the finding to be meaningful, because the result of this regimen, even without bevacizumab, was comparable to that reported in the IMELDA trial [16], and better than those reported in the AROBASE study [21].

### 3.2. Efficacy of Bevacizumab–Paclitaxel Reinduction Therapy

In 54 of 80 patients who were successfully switched to bevacizumab–paclitaxel reinduction therapy, PFS was 7.8 months. Added to the 6 months of PFS under induction therapy, the total treatment duration for bevacizumab–paclitaxel was >1 year, a similar length of time to the PFS achieved in the E2100 [11] and MERiDiAN trials [10,12]. Of the patients who did not receive reinduction therapy, about half chose endocrine therapy or other anticancer therapies. This suggests that they wished to avoid AEs such as peripheral neuropathy and alopecia, which they might have experienced while undergoing induction therapy or the symptom improved during maintenance therapy.

### 3.3. Effects on OS from the Start of Induction Therapy

The addition of capecitabine to endocrine maintenance therapy improved the OS rate (increase in point estimate by 21.2% and 15% at 24 and 36 months, respectively) from the start of induction therapy, consistent with the finding in the IMELDA study [16], suggesting that the increased PFS during maintenance therapy could lead to a positive effect on OS. The effect of capecitabine after bevacizumab–paclitaxel induction therapy may be explained by the mechanism discussed in the CREATE-X trial report; taxanes may induce thymidine phosphorylase in cancer cells, which activates capecitabine, resulting in increased antitumor effect [17,24].

The finding inversely suggests that endocrine-only therapy may not be sufficient at the maintenance stage after induction therapy. However, it should be noted that, in the present study, nearly 30% of the randomized patients had received ≥3 endocrine therapy regimens for AMBC at baseline. Therefore, in this population of patients, the benefits of maintenance endocrine therapy at this point may be suboptimal, potentially leading to the observed short PFS in group E compared with group EC. This is also supported by the finding from the ad hoc subgroup analysis showing previous use of ≥3 regimens of endocrine therapy as a factor affecting the PFS of maintenance therapy (Appendix A) and OS (Appendix A) in favor of group EC. Although limited by the small sample size, the results may also be interpreted as follows: the necessity of ≥3 regimens of endocrine therapy may indicate that tumors were more sensitive to endocrine therapy at baseline; however, the sensitivity may be weakened because of the long-term endocrine therapy. Regarding poststudy treatment, we did not collect relevant data (i.e., treatment details after completion of maintenance therapy and after completion of bevacizumab–paclitaxel reinduction therapy) through the electronic data capture system. However, as capecitabine is recommended as a standard treatment for AMBC by the guidelines and it is reimbursed by Japanese health insurance, it is possible that many patients in group E received treatment with capecitabine upon progression.

Since CDK4/6 inhibitors have now become available, the combination of endocrine therapy and a CDK4/6 inhibitor may be a possible option as maintenance therapy. Future study will be needed to determine whether capecitabine or a CDK4/6 inhibitor is more appropriate to be added to maintenance endocrine therapy in terms of prolonging OS.

### 3.4. Implications for Therapeutic Strategy

With the introduction of CDK4/6 inhibitors, endocrine therapy plus a CDK4/6 inhibitor is used as standard first- or second-line therapy for HR-positive, HER2-negative AMBC; it has demonstrated clinical benefit in terms of improved PFS with less toxicity as compared with a chemotherapy regimen [25,26,27]. However, the meta-analysis of data from the MONALEESA trials found differences in the efficacy of concomitant CDK4/6 inhibitors when patients with HR-positive, HER2-negative AMBC were stratified by a PAM50-based subtype [28]. For example, patients with the basal-like subtype are unlikely to benefit from the use of CDK4/6 inhibitors. For these patients, as well as those who had progression after treatment with CDK4/6 inhibitors, chemotherapies should be considered at an early treatment phase to eliminate the highly malignant tumors, and then followed by maintenance therapy that is less toxic. This concept was suggested by the IMELDA trial [16], and supported by the present study findings, in which chemotherapy-containing maintenance therapy was beneficial in terms of both PFS and OS.

### 3.5. Limitations and Future Research

The present phase 2 trial was not designed to compare different combinations of endocrine agents and capecitabine. Additionally, it was not designed to investigate whether alternatives to capecitabine, such as CDK4/6 inhibitors, would be efficacious in combination with endocrine maintenance therapy, or whether bevacizumab should be added to maintenance therapy. It was aimed at determining which of the maintenance therapies (endocrine therapy alone or capecitabine plus endocrine therapy) should move on to phase 3, which will then compare the standard treatment (continuing initial chemotherapy until reaching PD) and chemotherapy followed by maintenance therapy.

As CDK4/6 inhibitors had not been approved in Japan during the present study period, no patients treated with CDK4/6 inhibitors were included. The combination of endocrine therapy and a CDK4/6 inhibitor represents the recent standard first- and second-line treatment. However, as there has been no evidence suggesting that the use of a CDK4/6 inhibitor affects the sensitivity to subsequent chemotherapy, we believe that data from the present study evaluating the usefulness of adding capecitabine to endocrine therapy can also be extrapolated to patients previously treated with CDK4/6 inhibitors, given that they are still in need of chemotherapy.

About one-third of the patients (*n* = 13 each) in groups E and EC did not receive reinduction therapy. This may have affected the OS, given that the PFS of the reinduction therapy tended to be longer in group E than in group EC, as opposed to the result of the PFS of maintenance therapy.

As this was a phase 2 study, a limited number of patients were enrolled. However, patients were randomly assigned into the groups so as to minimize the effects of confounding factors. Moreover, the results of the subgroup analyses showed straightforward results; consistent superiority of group EC in terms of PFS (Appendix A) and OS (Appendix A) was shown. Therefore, we consider that discrepancies between the groups unlikely affected the present study findings. A meta-analysis of data and a phase 3 study are needed to confirm the efficacy of the treatment strategy comprising induction, maintenance, and reinduction therapy in AMBC patients and establish optimal therapeutic strategies. Additionally, translational research analysis of serum samples collected at each evaluation point is still ongoing; the results will be presented in due course.

## 4. Materials and Methods

### 4.1. Study Design

The present study, KBCSG-TR1214 (UMIN Clinical Trials Registry trial no. UMIN000008662), was a multicenter, parallel-group, open-label, randomized phase 2 trial for HR–positive, HER2-negative patients who had undergone no or one prior chemotherapy for AMBC. The study design is shown in Figure 6.

### 4.2. Recruitment

Patients were recruited from hospitals affiliated with Kinki Breast Cancer Study Group–Translational Research in Japan and registered at the Data Coordinating Center, Osaka University Hospital, Osaka, Japan. Eligible patients were women aged ≥20 years with advanced breast cancer with distant metastases or inoperable recurrent breast cancer who were expected to benefit from endocrine therapy. The other key inclusion criteria at primary registration were estrogen-receptor–positive status confirmed immunohistochemically, HER2-negative status confirmed by fluorescence in situ hybridization or immunohistochemically, Eastern Cooperative Oncology Group performance status of 0 or 1, life expectancy ≥12 months, evaluable lesion (according to the revised Response Evaluation Criteria in Solid Tumors (RECIST) guideline, version 1.1), previous treatment with no more than one chemotherapy regimen for metastatic or inoperable recurrent breast cancer, no previous treatment with paclitaxel (including nab-paclitaxel), oral fluoropyrimidines (with the exception of administration in a neoadjuvant or adjuvant setting, or ≥12 months before primary registration) or bevacizumab, and adequate organ function (confirmed by laboratory testing within 2 weeks before receiving the first dose of any study drug).

The key exclusion criteria at primary registration included brain metastases, pregnancy, and ineligibility based on the decision of the attending physician or the principal investigator.

The key inclusion criteria at secondary registration included CR, PR, or stable disease (SD), confirmed by imaging at the end of cycle 6 of induction therapy; CR, PR, or SD (confirmed by imaging) in case of discontinuation after ≥4 cycles of induction therapy due to AEs; no obvious sign of PD (confirmed by imaging at the end of induction therapy), and no clinical worsening, in patients with a small lesion (maximum diameter, <10 mm, or minimum lymph node diameter, ≥10 mm and <15 mm) only, bone lesions only, pleural lesions only, or diffuse skin lesions only (this criterion also applied to patients who received ≥4 cycles of induction therapy but discontinued due to AEs); and adequate organ function (confirmed by laboratory testing).

### 4.3. Treatment

#### 4.3.1. Induction Therapy

The initial treatment was induction therapy comprising 6 cycles of bevacizumab and paclitaxel. Each cycle was 28 days. Patients received bevacizumab (10 mg/kg on days 1 and 15) and paclitaxel (90 mg/m^2^ on days 1, 8, and 15). During induction therapy, concomitant use of other anticancer therapies, or of antithrombotic agents (other than aspirin ≤324 mg/day), was prohibited.

#### 4.3.2. Randomization and Maintenance Therapy

After 6 cycles (≥4 cycles in case of discontinuation due to AEs) of induction therapy, patients without PD were randomly assigned in a 1:1 ratio to receive maintenance therapy with either endocrine therapy alone (group E) or endocrine therapy plus capecitabine (group EC). The stratification factors used at randomization were menopausal status, presence of a measurable lesion, institution, number of prior endocrine therapies for AMBC (<3 or ≥3), and whether or not the treatment was first-line chemotherapy for AMBC.

In both group E and group EC, the choice of endocrine agents was at the discretion of the attending physicians. The drugs were administered in accordance with the package inserts. In group EC, patients additionally received capecitabine, 1657 mg/m^2^/day twice daily on days 1–21, followed by a 7-day rest period. This metronomic regimen of capecitabine was based on previous studies indicating a less toxic profile associated with low-dose capecitabine [29,30]. Prohibited treatments during maintenance therapy were the same as those during induction therapy. Use of LHRH agonists (goserelin and leuprorelin) was permitted during maintenance therapy.

#### 4.3.3. Reinduction Therapy

In cases of PD despite maintenance therapy, patients were switched to bevacizumab–paclitaxel reinduction therapy. In cases of suspension or discontinuation of either drug (i.e., bevacizumab or paclitaxel), administration of the other continued. If neither drug could be administered within 28 (+3) days of the most recent dose, the reinduction therapy was judged to be terminated.

### 4.4. Assessment of Efficacy and Safety

Efficacy was assessed according to the revised RECIST guideline, version 1.1 [31]. Adverse events were defined as any unfavorable or unintended sign that may or may not be associated with the study treatment. Periodic monitoring was carried out, and all AEs were recorded and graded in accordance with the National Cancer Institute Common Terminology Criteria for Adverse Events, version 4.0 (Japanese Clinical Oncology Group edition) [32].

Follow-up was for 3 years after completion of registration, or for patients registered after August 2015, the end of treatment.

### 4.5. Endpoints

The primary endpoint was PFS of maintenance therapy (defined as the period from the secondary registration to the first of either PD or death from any cause). The main secondary endpoints were time to failure of strategy (defined as the period from the secondary registration to the first of either PD during the reinduction therapy, death from any cause, or date of treatment discontinuation—in the case of patients who did not receive reinduction therapy, date of PD during maintenance therapy); ORR (defined as proportion of patients with CR or PR) and DCR (defined as proportion of patients with CR, PR, or SD) of reintroduction therapy; PFS of reinduction therapy (defined as the period from the start of reinduction therapy to the first of either PD or death from any cause) and time to treatment failure of reintroduction therapy (defined as the period from the start of reinduction therapy to the first of either PD, death from any cause, or date of treatment discontinuation); OS (defined as the period from the primary registration to death from any cause); and safety.

### 4.6. Statistical Analyses

Sample size calculations were calculated based on a type 1 error (one sided) of 0.05, an 80% power to estimate the median PFS in patients who had received maintenance therapy (endocrine therapy with or without capecitabine) for 9 months (threshold 6 months), a 2-year enrollment period, and a 1-year follow-up period, and accounted for the exclusion of ineligible patients. The target number of patients enrolled and the target number randomized after induction therapy (having completed primary registration) were 120 and 90, respectively.

Data used in the efficacy analysis were from the full analysis set, defined as patients who fulfilled the inclusion criteria and were enrolled to receive study treatment. The safety analysis used data from all patients who received at least one dose of any of the maintenance therapy drugs. The Kaplan–Meier method was used to generate curves for PFS and other survival endpoints, and to calculate related statistics, for groups E and EC.

The trial was designed not to directly compare the results for groups E and EC but rather to determine whether the survival endpoint results for each group met prespecified criteria: threshold median PFS ≥6.0 months and expected median PFS ≥9 months. By means of comprehensive assessment of the results for these and other endpoints (including incidence of AEs), we aimed to determine which therapies should move on to phase 3. Both groups E and EC were experimental arms. Therefore, the following comparisons were made for the purpose of reference only. HRs and their 95% CIs were calculated by Cox regression analysis, and intergroup comparisons were performed by the log-rank test. Based on the Cox model, ad hoc subgroup analyses using stratification factors at randomization were carried out to identify factors affecting survival. Regarding the response rate (ORR and DCR), Fisher’s exact test was used to compare the results between groups E and EC. Statistical analyses were carried out using SPSS Statistics 26.0 (IBM Corp., Armonk, NY, USA) and R (version 3.5.2) [33].

## 5. Conclusions

We found that bevacizumab–paclitaxel induction therapy followed by maintenance endocrine therapy and subsequent bevacizumab–paclitaxel reinduction therapy could be a new therapeutic strategy for HR-positive, HER2-negative AMBC. Our results suggest that the addition of capecitabine to endocrine therapy may be more beneficial than endocrine therapy alone as maintenance therapy.

## Figures and Tables

**Figure 1 cancers-13-04399-f001:**
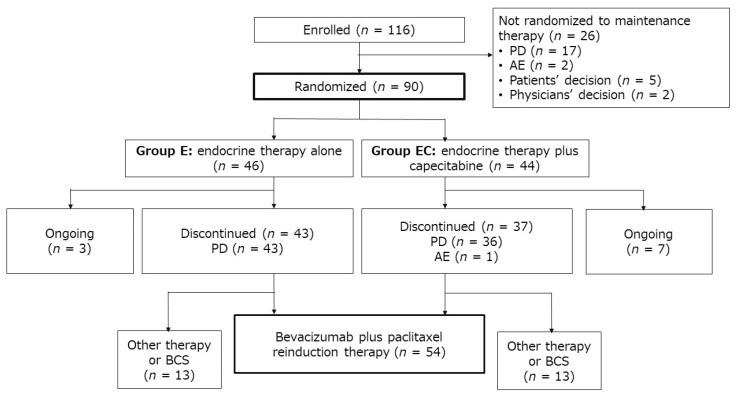
Patient disposition. AE, adverse event; BCS, breast-conserving surgery; E, endocrine therapy alone; EC, endocrine therapy plus capecitabine; PD, progressive disease.

**Figure 2 cancers-13-04399-f002:**
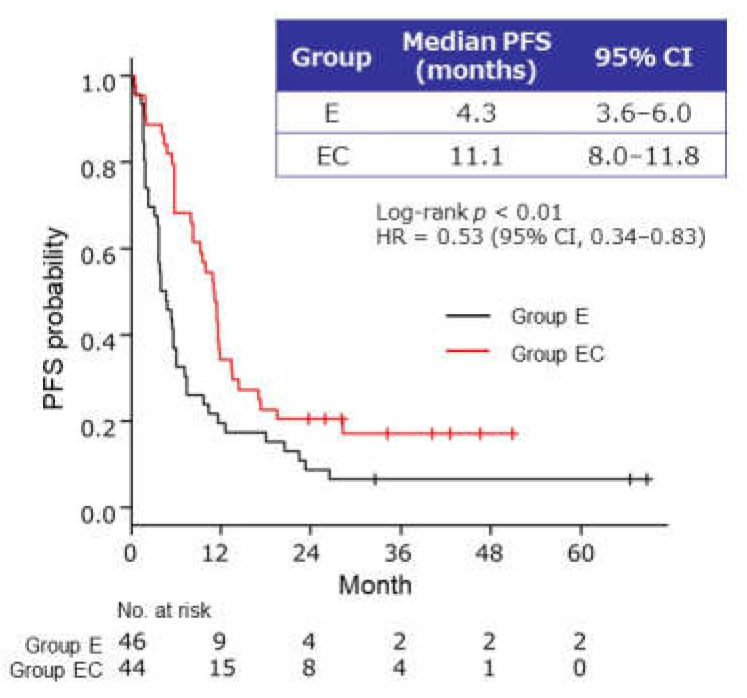
Kaplan–Meier curves of progression-free survival (PFS) in patients who received maintenance therapy with endocrine therapy alone (group E) or endocrine therapy plus capecitabine (group EC) after bevacizumab–paclitaxel induction therapy. CI, confidence interval; E, endocrine therapy alone; EC, endocrine therapy plus capecitabine; HR, hazard ratio.

**Figure 3 cancers-13-04399-f003:**
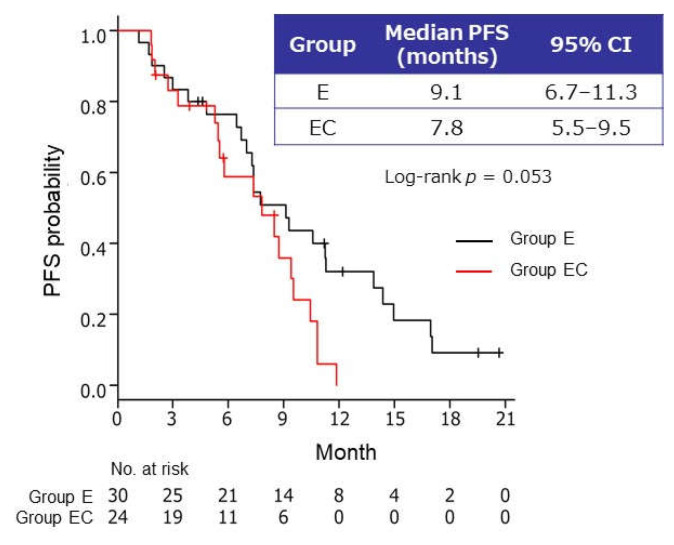
Kaplan–Meier curves of progression-free survival (PFS) in patients who received bevacizumab–paclitaxel reinduction therapy after maintenance therapy with endocrine therapy alone (group E) or endocrine therapy plus capecitabine (group EC). CI, confidence interval; E, endocrine therapy alone; EC, endocrine therapy plus capecitabine.

**Figure 4 cancers-13-04399-f004:**
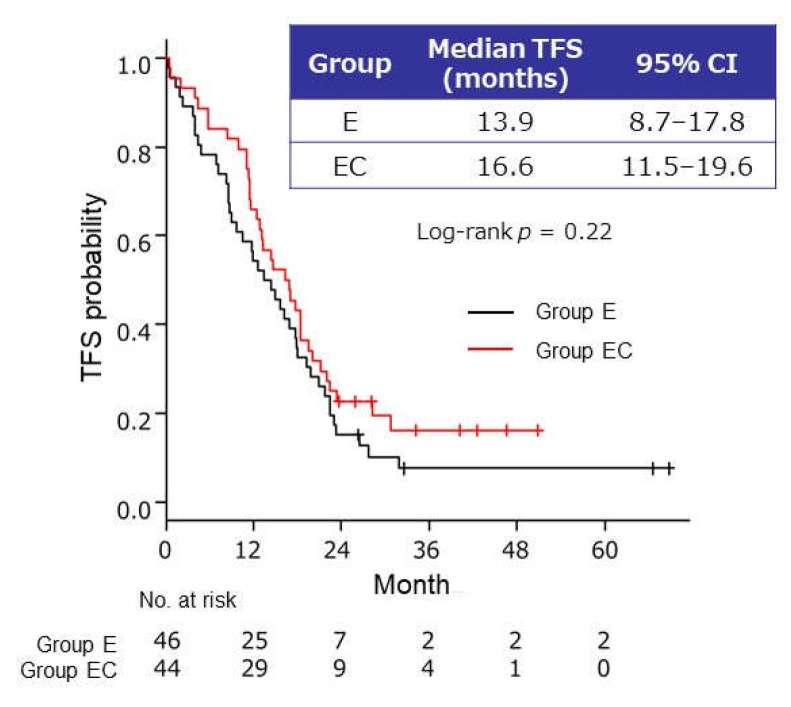
Kaplan–Meier curves of time to failure of strategy (TFS) from randomization in patients who received bevacizumab–paclitaxel reinduction therapy after maintenance therapy with endocrine therapy alone (group E) or endocrine therapy plus capecitabine (group EC). CI, confidence interval; E, endocrine therapy alone; EC, endocrine therapy plus capecitabine.

**Figure 5 cancers-13-04399-f005:**
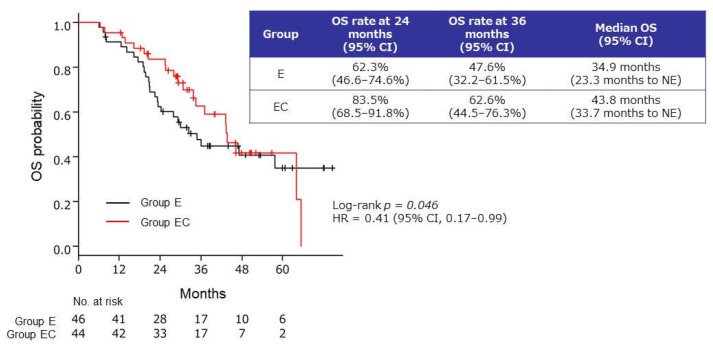
Kaplan–Meier curves of overall survival (OS) rate from the start of bevacizumab–paclitaxel induction therapy in patients who received maintenance therapy with endocrine therapy alone (group E) or endocrine therapy plus capecitabine (group EC). CI, confidence interval; E, endocrine therapy alone; EC, endocrine therapy plus capecitabine; HR, hazard ratio; NE, not estimable.

**Figure 6 cancers-13-04399-f006:**
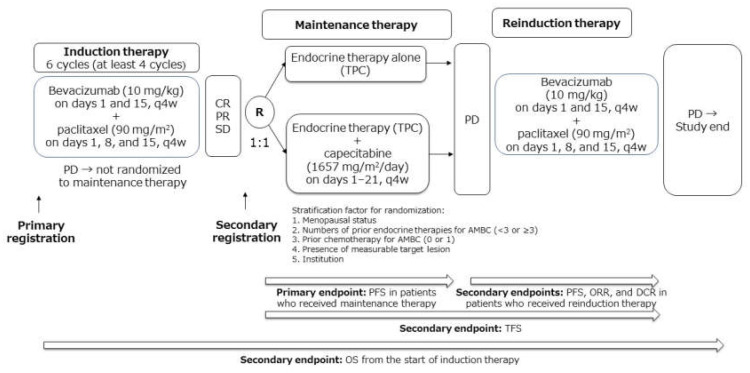
Study design. AMBC, advanced and metastatic breast cancer; DCR, disease control rate; CR, complete response; ORR, objective response rate; PD, progressive disease; PFS, progression-free survival; PR, partial response; R, randomization; SD, stable disease; TFS, time from randomization to failure of strategy; TPC, treatment of physician’s choice; q4w, every 4 weeks.

**Table 1 cancers-13-04399-t001:** Baseline characteristics.

Characteristic	Enrolled (*n* = 116)	Randomized (*n* = 90)	Group E (*n* = 46)	Group EC (*n* = 44)
Median Age, Years (Range)	59.8 (31.5–81.0)	59.8 (34.5–81.0)	59.8 (41.5–75.8)	59.9 (34.5–81.0)
Menopausal Status				
Premenopausal	26 (22.4)	20 (22.2)	10 (21.7)	10 (22.7)
Postmenopausal	90 (77.6)	70 (77.8)	36 (78.3)	34 (77.3)
ECOG PS				
0	83 (71.6)	64 (71.1)	33 (71.7)	31 (70.5)
1	32 (27.6)	25 (27.8)	13 (28.3)	12 (27.3)
2	1 (0.9)	1 (1.1)	0	1 (02.3)
Measurable Lesion ^a^				
Yes	111 (95.7)	85 (94.4)	44 (95.7)	41 (93.2)
No	5 (4.3)	5 (5.6)	2 (4.3)	3 (6.8)
Previous Endocrine Therapies with/without Targeted Therapy for AMBC				
0	42 (36.2)	35 (38.9)	18 (39.1)	17 (38.6)
1 Regimen	26 (22.4)	13 (14.4)	7 (15.2)	6 (13.6)
2 Regimens	19 (16.4)	16 (17.8)	8 (17.4)	8 (18.2)
≥3 Regimens	29 (25.0)	26 (28.9)	13 (28.3)	13 (29.5)
Exemestane + mTOR Inhibitor	8 (6.9)	7 (7.8)	4 (8.7)	3 (6.8)
Previous Chemotherapy for AMBC				
0	102 (87.9)	81 (90.0)	39 (84.8)	42 (95.5)
1 Regimen	14 (12.1)	9 (10.0)	7 (15.2)	2 (4.5)
Histological Grade				
1	19 (16.4)	15 (16.7)	8 (17.4)	7 (15.9)
2	41 (35.3)	32 (35.6)	17 (37.0)	15 (34.1)
3	23 (19.8)	15 (16.7)	6 (13.0)	9 (20.5)
Unknown	33 (28.4)	28 (31.1)	15 (32.6)	13 (29.5)
Hormone Receptor Status				
ER + PgR+	96 (82.8)	75 (83.3)	38 (82.6)	37 (84.1)
ER + PgR−	18 (15.5)	13 (14.4)	7 (15.2)	6 (13.6)
ER + PgR Unknown	2 (1.7)	2 (2.2)	1 (2.2)	1 (2.3)
HER2 Status (IHC)				
0	63 (54.3)	48 (53.3)	26 (56.5)	22 (50.0)
1+	40 (34.5)	32 (35.6)	16 (33.8)	16 (36.4)
2+	11 (9.5)	8 (8.9)	4 (8.7)	4 (9.1)
Unknown	2 (1.7)	2 (2.2)	0	2 (4.5)

Data expressed as *n* (%) unless otherwise specified. ^a^ according to the revised Response Evaluation Criteria in Solid Tumors guideline, version 1.1; AMBC, advanced and metastatic breast cancer; E, endocrine therapy alone; EC, endocrine therapy plus capecitabine; ECOG PS, Eastern Cooperative Oncology Group performance status; ER, estrogen receptor; IHC, immunohistochemistry; mTOR, mammalian target of rapamycin; PgR, progesterone receptor.

**Table 2 cancers-13-04399-t002:** Endocrine agents used as maintenance therapy.

Endocrine Agent(s)	Group E (*n* = 46)	Group EC (*n* = 44)
AI ± LHRH Agonist	28 (60.9)	27 (61.4)
SERM	11 (23.9)	8 (18.2)
SERD (Fulvestrant)	6 (13.0)	6 (13.6)
MPA	1 (2.2)	3 (6.8)

Data expressed as *n* (%). AI, aromatase inhibitor; E, endocrine therapy alone; EC, endocrine therapy plus capecitabine; LHRH, luteinizing-hormone-releasing hormone; MPA, medroxyprogesterone acetate; SERD, selective estrogen receptor downregulator; SERM, selective estrogen receptor modulator (tamoxifen, toremifene).

**Table 3 cancers-13-04399-t003:** Response to bevacizumab–paclitaxel induction therapy.

Response	Group E (*n* = 46)	Group EC (*n* = 44)
Complete Response	0	1 (2.3)
Partial Response	35 (76.1)	33 (75.0)
Stable Disease	11 (23.9)	10 (22.7)
Progressive Disease	0	0
Not Evaluable	0	0

Data expressed as *n* (%). E, endocrine therapy alone; EC, endocrine therapy plus capecitabine.

**Table 4 cancers-13-04399-t004:** Incidence of adverse events (experienced by ≥10% of patients or grade ≥3 in either group) in patients receiving endocrine therapy alone (group E) or endocrine therapy plus capecitabine (group EC) as maintenance therapy after bevacizumab–paclitaxel induction therapy.

Adverse Event	Group E (*n* = 46)		Group EC (*n* = 44)	
	*n*	Grade ≥ 3	*n*	Grade ≥ 3
Sensory Neuropathy	31 (67.4)	2 (4.3)	34 (77.3)	2 (4.5)
Alopecia	30 (65.2)	0 (0.0)	30 (68.2)	0 (0.0)
Hypertension	16 (34.8)	3 (6.5)	21 (47.7)	5 (11.4)
Hemorrhage	9 (19.6)	0 (0.0)	11 (25.0)	0 (0.0)
Proteinuria	9 (19.6)	0 (0.0)	4 (9.1)	1 (2.3)
Anemia	8 (17.4)	0 (0.0)	9 (20.5)	0 (0.0)
Malaise	7 (15.2)	0 (0.0)	14 (31.8)	0 (0.0)
Motor Neuropathy	7 (15.2)	0 (0.0)	8 (18.2)	0 (0.0)
Arthralgia	7 (15.2)	1 (2.2)	7 (15.9)	0 (0.0)
Anorexia	6 (13.0)	1 (2.2)	14 (31.8)	0 (0.0)
ALT Increased	6 (13.0)	2 (4.3)	1 (2.3)	0 (0.0)
Rash, Skin Changes (Dryness/Itchiness)	5 (10.9)	0 (0.0)	12 (27.3)	0 (0.0)
Limb Edema	5 (10.9)	0 (0.0)	9 (20.5)	0 (0.0)
Oral Mucositis	5 (10.9)	0 (0.0)	8 (18.2)	0 (0.0)
Fatigue	4 (8.7)	1 (2.2)	17 (38.6)	1 (2.3)
Nausea	4 (8.7)	0 (0.0)	10 (22.7)	0 (0.0)
AST Increased	4 (8.7)	2 (4.3)	1 (2.3)	0 (0.0)
Dysgeusia	3 (6.5)	0 (0.0)	12 (27.3)	0 (0.0)
Diarrhea	3 (6.5)	0 (0.0)	7 (15.9)	0 (0.0)
Muscle Pain	2 (4.3)	0 (0.0)	6 (13.6)	0 (0.0)
Palmar–Plantar Erythrodysesthesia Syndrome	1 (2.2)	0 (0)	33 (75.0)	6 (13.6)
Neutropenia	1 (2.2)	0 (0.0)	10 (22.7)	1 (2.3)
Leukopenia	1 (2.2)	0 (0.0)	7 (15.9)	0 (0.0)
Nail Disorders	0 (0.0)	0 (0.0)	13 (29.5)	0 (0.0)
CPK Increased	0 (0.0)	0 (0.0)	1 (2.3)	1 (2.3)

Data expressed as *n* (%). ALT, alanine aminotransferase; AST, aspartate aminotransferase; CPK, creatine phosphokinase; E, endocrine therapy alone; EC, endocrine therapy plus capecitabine.

## Data Availability

Data are available on request due to privacy/ethical restrictions.

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
