# Peer review of "Capecitabine in Combination with Endocrine Therapy as Maintenance Therapy after Bevacizumab Plus Paclitaxel Induction Therapy for Hormone Receptor-Positive, HER2-Negative Metastatic Breast Cancer: KBCSG-TR1214"

_cancers, 2021, doi:10.3390/cancers13174399_

Round 1

Reviewer 1 Report

This randomised trial preliminarily evaluated the benefit of adding capecitabine to maintenance endocrine therapy after an induction chemotherapy with Paclitaxel + Bevacizumab in HR+/HER2-negative MBC. Interesting results suggest that this option might improve PFS and OS. However I have many major and minor concerns with this study, mostly in the way it was presented, but partially also in the way it was conceived. Here are the following: 

1) Can the authors explain why they did not want to formally compare the two treatment arms? If the study was maybe intended as a phase II, why did they not use an activity endpoint to allow for a formal comparison between the two arms? For example the overall response rate or the disease control rate. 

2) The authors state that subgroup analyses were preplanned. However there is no formal preplanned statistical comparison between the two treatment arms and, consequently, there was no control of the alfa error for multiplicity for the subgroup analyses. All these analyses presented are substantially hypothesis-generating and the authors should far better stress this. 

3) Did all patients previously received an upfront endocrine therapy +/- target therapy? How many patients received a CDK4/6 inhibitor in first or second line? Which are the most appropriate setting where to administer such drugs. And did any receive Everolimus+Exemestane? Fulvestrant +/- alpelisib in PIK3CA mutant cases?

Previous ET should be overall better defined, since every guideline recommend to treat with as many ET as possible all HR+/HER2-neg. MBC, unless in case of visceral crisis. The author should better stress this aspect and cite results from the following studies supporting this concept (PMID 31494037, 31668850 and 33385521, at least). 

5) Following the previous point, lines 368-370: I would suggest the authors to be more careful in their evaluation of maintenance ET. These patients were already ET-pretreated, meaning that the ET adopted for maintenance might be suboptimal in this context, but not necessarily in a context where the CT is used as upfront treatment (e.g. visceral crisis) and is then suspended to administer a first-line ET+CDK4/6 inhibitor, for example. Therefore, more caution should be used in that statement and a better contextualisation. 

5) Patients experiencing PD within 6 cycles and changing treatment line for this cause are a completely different population from patients receiving a predefined number of CT cycles switching to a maintenance treatment. Is obvious that the first should perform worse. In this perspective, the analyses comparing this small enrolled population that could not be randomised to maintenance, with the one randomised, is quite misleading. It would be also important to know the subsequent treatment line used for progressing patients, which was compared to ET+cape/ET maintenance. All this section might be better moved to supplementary materials, because is out of the main study focus. 

6) The study was not designed to formally compare the PFS difference between the 2 arms, therefore it was not sufficiently powered to draw any meaningful conclusion. A phase 3 comparative trial, possibly double blinded and with a statistical plan developed to assess the difference in PFS in terms of hazard ratio and 95%CI should be run to offer more solid results. The present study is however interesting and generate the hypothesis to run such trial. However, the author should downsize the overall tone of the Discussion, be more careful in the conclusion that they draw and be more realistic on the current clinical applicability of this trial. 

Author Response

Reviewer 1

This randomised trial preliminarily evaluated the benefit of adding capecitabine to maintenance endocrine therapy after an induction chemotherapy with Paclitaxel + Bevacizumab in HR+/HER2-negative MBC. Interesting results suggest that this option might improve PFS and OS. However I have many major and minor concerns with this study, mostly in the way it was presented, but partially also in the way it was conceived. Here are the following: 

1) Can the authors explain why they did not want to formally compare the two treatment arms? If the study was maybe intended as a phase II, why did they not use an activity endpoint to allow for a formal comparison between the two arms? For example the overall response rate or the disease control rate. 

Response

Our primary aims in this study were to evaluate which maintenance therapy, i.e. endocrine therapy alone or capecitabine plus endocrine therapy, is more useful with less toxicity in patients whose disease was controlled after receiving bevacizumab–paclitaxel induction therapy, and also to investigate whether or not bevacizumab–paclitaxel reinduction therapy after maintenance therapy is feasible. As we believe that the duration of efficacy of the studied regimen is the most important endpoint in the treatment of advanced recurrent breast cancer, we chose PFS under maintenance therapy as a primary endpoint.  

Although, as you pointed, overall response rate (ORR) and disease control rate (DCR) are frequently-used endpoints in a phase 2 study, in the present study however, it is likely that they may be affected by the effects of induction therapy. In fact, while over 90% of patients had measurable lesions at baseline, ≥75% of the patients achieved PR after bevacizumab–paclitaxel induction therapy, for whom tumor size determination can be difficult. However, we also believe that ORR and DCR are important endpoints, and therefore we analyzed them as secondary endpoints.

2) The authors state that subgroup analyses were preplanned. However there is no formal preplanned statistical comparison between the two treatment arms and, consequently, there was no control of the alfa error for multiplicity for the subgroup analyses. All these analyses presented are substantially hypothesis-generating and the authors should far better stress this. 

Response

As you pointed out, in the present randomized phase 2 trial, we aimed to determine which of the maintenance therapies (endocrine therapy alone or capecitabine plus endocrine therapy) should move on to phase 3, which will then compare standard treatment (continuing bevacizumab–paclitaxel therapy until reaching PD) and bevacizumab–paclitaxel therapy followed by maintenance therapy. Therefore, both groups E and EC are experimental arms.

We have clarified this in the Statistical Analyses section.

Since the subgroup analyses provide complementary data for interpreting the present study results, we have moved Figure 3 and Figure 7 to Supplementary Materials, and revised the main text accordingly.

3) Did all patients previously received an upfront endocrine therapy +/- target therapy? How many patients received a CDK4/6 inhibitor in first or second line? Which are the most appropriate setting where to administer such drugs. And did any receive Everolimus+Exemestane? Fulvestrant +/- alpelisib in PIK3CA mutant cases?

Response

In Table 1, we have included the detail list of previous endocrine therapy with/without targeted therapy for AMBC. As CDK4/6 inhibitors had not been approved in Japan during the study period and alpelisib to date has not yet been approved, no patients were treated with these drugs. As shown in Table 1, only 7.8% (7/90) of the randomized patients received everolimus + exemestane; the patients in the present study were registered between 2012 and 2016, but mTOR inhibitor was approved in Japan only from 2014.

As you pointed out, the recent standard first- and secondary-line endocrine therapy for AMBC includes endocrine therapy plus CDK4/6 inhibitor. Because CDK4/6 inhibitors had not been approved in Japan during the present study period, no patients treated with CDK4/6 inhibitors were included. However, there has been no evidence suggesting that the use of CDK4/6 inhibitor affects the sensitivity to subsequent chemotherapy; prolonged OS has been confirmed in patients treated with CDK4/6 inhibitors. Therefore, we believe that data from the present study evaluating the usefulness of adding capecitabine to endocrine therapy can also be extrapolated to patients previously treated with CDK4/6 inhibitors, given that they are still in need of chemotherapy.

We have included the above interpretation under the Limitations and future research.

Previous ET should be overall better defined, since every guideline recommend to treat with as many ET as possible all HR+/HER2-neg. MBC, unless in case of visceral crisis. The author should better stress this aspect and cite results from the following studies supporting this concept (PMID 31494037, 31668850 and 33385521, at least). 

Response

Thank you for your recommendation. In accordance with your suggestion, we have revised the Introduction and included citations for these additional references.

New references (numbers 8, 9, 10 in the revised manuscript):

l  Giuliano, M.; Schettini, F.; Rognoni, C.; Milani, M.; Jerusalem, G.; Bachelot, T.; De Laurentiis, M.; Thomas, G.; De Placido, P.; Arpino, G.; De Placido, S.; Cristofanilli, M.; Giordano, A.; Puglisi, F.; Pistilli, B.; Prat, A.; Del Mastro, L.; Venturini, S.; Generali, D. Endocrine treatment versus chemotherapy in postmenopausal women with hormone receptor-positive, HER2-negative, metastatic breast cancer: a systematic review and network meta-analysis. Lancet. Oncol. 2019, 20, 1360-1369.

l  Park, Y.H.; Kim, T.Y.; Kim, G.M.; Kang, S.Y.; Park, I.H.; Kim, J.H.; Lee, K.E.; Ahn, H.K.; Lee, M.H.; Kim, H.J.; Kim, H.J.; Lee, J.I.; Koh, S.J.; Kim, J.Y.; Lee, K.H.; Sohn, J.; Kim, S.B.; Ahn, J.S.; Im, Y.H.; Jung, K.H.; Im, S.A.; Korean Cancer Study Group (KCSG). Palbociclib plus exemestane with gonadotropin-releasing hormone agonist versus capecitabine in premenopausal women with hormone receptor-positive, HER2-negative metastatic breast cancer (KCSG-BR15-10): a multicentre, open-label, randomised, phase 2 trial. Lancet. Oncol. 2019, 20, 1750-1759.

l  Martin, M.; Zielinski, C.; Ruiz-Borrego, M.; Carrasco, E.; Turner, N.; Ciruelos, E.M.; Muñoz, M.; Bermejo, B.; Margeli, M.; Anton, A.; Kahan, Z.; Csöszi, T.; Casas, M.I.; Murillo, L.; Morales, S.; Alba, E.; Gal-Yam, E.; Guerrero-Zotano, A.; Calvo, L.; de la Haba-Rodriguez, J.; Ramos, M.; Alvarez, I.; Garcia-Palomo, A.; Huang Bartlett, C.; Koehler, M.; Caballero, R.; Corsaro, M.; Huang, X.; Garcia-Sáenz, J.A.; Chacón, J.I.; Swift, C.; Thallinger, C.; Gil-Gil, M. Palbociclib in combination with endocrine therapy versus capecitabine in hormonal receptor-positive, human epidermal growth factor 2-negative, aromatase inhibitor-resistant metastatic breast cancer: a phase III randomised controlled trial-PEARL. Ann. Oncol. 2021, 32, 488-499.

5) Following the previous point, lines 368-370: I would suggest the authors to be more careful in their evaluation of maintenance ET. These patients were already ET-pretreated, meaning that the ET adopted for maintenance might be suboptimal in this context, but not necessarily in a context where the CT is used as upfront treatment (e.g. visceral crisis) and is then suspended to administer a first-line ET+CDK4/6 inhibitor, for example. Therefore, more caution should be used in that statement and a better contextualisation. 

Response

We have revised the relevant part under 3.3. Effects on OS from the start of induction therapy in the Discussion as follows:

Also, it inversely suggests that endocrine-only therapy may not be sufficient at the maintenance stage after induction therapy. However, it should be noted that, in the present study, nearly 30% of the randomized patients had received ≥3 endocrine therapy regimens for AMBC at baseline. Therefore, in this population of patients, the benefits of maintenance endocrine therapy at this point may be suboptimal, potentially leading to the observed short PFS in group E compared to group EC. This is also supported by the finding from the ad-hoc subgroup analysis showing previous use of ≥ 3 regimens of endocrine therapy as a factor affecting PFS of maintenance therapy in favor of group EC. Since CDK4/6 inhibitors have now become available, the combination of endocrine therapy and CDK4/6 inhibitor may be a possible option as maintenance therapy.

5) Patients experiencing PD within 6 cycles and changing treatment line for this cause are a completely different population from patients receiving a predefined number of CT cycles switching to a maintenance treatment. Is obvious that the first should perform worse. In this perspective, the analyses comparing this small enrolled population that could not be randomised to maintenance, with the one randomised, is quite misleading. It would be also important to know the subsequent treatment line used for progressing patients, which was compared to ET+cape/ET maintenance. All this section might be better moved to supplementary materials, because is out of the main study focus. 

Response

As suggested, we have omitted Figure 8 and relevant results/discussion from the manuscript.

6) The study was not designed to formally compare the PFS difference between the 2 arms, therefore it was not sufficiently powered to draw any meaningful conclusion. A phase 3 comparative trial, possibly double blinded and with a statistical plan developed to assess the difference in PFS in terms of hazard ratio and 95%CI should be run to offer more solid results. The present study is however interesting and generate the hypothesis to run such trial. However, the author should downsize the overall tone of the Discussion, be more careful in the conclusion that they draw and be more realistic on the current clinical applicability of this trial. 

Response

As suggested, we have toned down the general expression for interpretation of the study finding, and revised the Limitations and Conclusions as follows.

3.5. Limitations and future research

“The present phase 2 trial was not designed to compare different combinations of endocrine agents and capecitabine. Also, it was not designed to investigate whether alternatives to capecitabine, such as CDK4/6 inhibitors, would be efficacious in combination with endocrine maintenance therapy, or whether bevacizumab should be added to maintenance therapy. It aimed to determine which of the maintenance therapies (endocrine therapy alone or capecitabine plus endocrine therapy) should move on to phase 3, which will then compare the standard treatment (continuing initial chemotherapy until reaching PD) and chemotherapy followed by maintenance therapy. As CDK4/6 inhibitors had not been approved in Japan during the present study period, no patients treated with CDK4/6 inhibitors were included. The combination of endocrine therapy and CDK4/6 inhibitor represent the recent standard first- and second-line treatment. However, as there has been no evidence suggesting that the use of CDK4/6 inhibitor affects the sensitivity to subsequent chemotherapy, we believe that data from the present study evaluating the usefulness of adding capecitabine to endocrine therapy can also be extrapolated to patients previously treated with CDK4/6 inhibitors, given that they are still in need of chemotherapy.”

Conclusions

“Our results suggest that the addition of capecitabine to maintenance endocrine therapy prolongs PFS and OS in patients with HR-positive, HER2-negative AMBC. To our knowledge, this is the first clinical trial to show the efficacy of endocrine therapy combined with metronomic capecitabine as maintenance therapy after induction chemotherapy in HR-positive, HER2-negative AMBC patients. Induction therapy followed by maintenance therapy could be a new therapeutic strategy for ABMC.”

Reviewer 2 Report

In this small randomized phase 2 trial the Authors compare two maintenance strategies following paclitaxel-bevacizumab induction chemotherapy (given as a first-line chemotherapy in 90% of patients) in HR-positive Her 2 negative metastatic breast cancer patients: endocrine therapy alone (a strategy followed by most Oncologists around the world)  or endocrine therapy  combined with a chemotherapeutic drug, capecitabine. The main conclusions are that the addition of chemotherapy to maintenance endocrine therapy improves PFS and OS with a manageable safety profile.

I have the following observations:

  1. The majority of enrolled patients had already received at least one or two lines of endocrine therapy for MBC (72.2%; it is not clear how many received 1 line and how many received 2 lines); the remaining had already received three or more lines. It is therefore clear that this is a population of patients for whom the benefits of endocrine therapy had been more or less fully exploited, as confirmed by the very short PFS observed in patients randomized to the E only arm and by the fact that the only factor associated with a significantly longer PFS in the EC arm in the Forest plot is observed in patients who had received ≥3 of previous line of endocrine therapy
  2. The Article is well written but is too heavy to follow, essentially because the Authors want to get too many results from a small group of patients. I suggest taking out figures number 4, 5, 7.
  3. The Authors detail the efficacy of paclitaxel-bevacizumab reinduction therapy but they do not mention the use of capecitabine upon progression in the E-only arm.
  4. Differences between cancer treatments frequently result from discrepancies between the groups receiving the treatment rather than from any superiority of one particular regimen over another. This is particularly true for trials involving fewer than 200 patients, in which significant imbalances in patients’ characteristics can lead to misleading conclusions.

Author Response

Reviewer 2

In this small randomized phase 2 trial the Authors compare two maintenance strategies following paclitaxel-bevacizumab induction chemotherapy (given as a first-line chemotherapy in 90% of patients) in HR-positive Her 2 negative metastatic breast cancer patients: endocrine therapy alone (a strategy followed by most Oncologists around the world)  or endocrine therapy  combined with a chemotherapeutic drug, capecitabine. The main conclusions are that the addition of chemotherapy to maintenance endocrine therapy improves PFS and OS with a manageable safety profile.

I have the following observations:

  1. The majority of enrolled patients had already received at least one or two lines of endocrine therapy for MBC (72.2%; it is not clear how many received 1 line and how many received 2 lines); the remaining had already received three or more lines. It is therefore clear that this is a population of patients for whom the benefits of endocrine therapy had been more or less fully exploited, as confirmed by the very short PFS observed in patients randomized to the E only arm and by the fact that the only factor associated with a significantly longer PFS in the EC arm in the Forest plot is observed in patients who had received ≥3 of previous line of endocrine therapy

Response

We have updated Table 1 to include the detailed list of previous endocrine therapy.

Also, we have revised the text under 3.3. Effects on OS from the start of induction therapy Discussion as follows:

“Also, it inversely suggests that endocrine-only therapy may not be sufficient at the maintenance stage after induction therapy. However, it should be noted that, in the present study, nearly 30% of the randomized patients had received ≥3 endocrine therapy regimens for AMBC at baseline. Therefore, in this population of patients, the benefits of maintenance endocrine therapy at this point may be suboptimal, potentially leading to the observed short PFS in group E compared to group EC. This is also supported by the finding from the ad-hoc subgroup analysis showing previous use of ≥ 3 regimens of endocrine therapy as a factor affecting PFS of maintenance therapy in favor of group EC. Since CDK4/6 inhibitors have now become available, the combination of endocrine therapy and CDK4/6 inhibitor may be a possible option as maintenance therapy.”

  1. The Article is well written but is too heavy to follow, essentially because the Authors want to get too many results from a small group of patients. I suggest taking out figures number 4, 5, 7.

Response

As also suggested by other reviewer, we have moved Figures 3 and 7 to Supplementary Materials. Also, we have deleted Figure 8. We wish to keep Figures 4 and 5 as they contain relevant secondary endpoint data.

  1. The Authors detail the efficacy of paclitaxel-bevacizumab reinduction therapy but they do not mention the use of capecitabine upon progression in the E-only arm.

Response

Unfortunately, we did not collect relevant data (i.e. post-study treatment details after completion of maintenance therapy and after completion of bevacizumab–paclitaxel reinduction therapy reinduction therapy) through the Electronic Data Capture system, thus, it is not clear how many of the patients in group E actually received capecitabine. However, as capecitabine is recommended as a standard treatment for AMBC by the guidelines and it is reimbursed by Japanese health insurance, it is possible that many patients in group E received treatment with capecitabine upon progression.

We have added the above statement in the Discussion under 3.3. Effects on OS from the start of induction therapy.

  1. Differences between cancer treatments frequently result from discrepancies between the groups receiving the treatment rather than from any superiority of one particular regimen over another. This is particularly true for trials involving fewer than 200 patients, in which significant imbalances in patients’ characteristics can lead to misleading conclusions.

Response

We appreciate your concern about the possibility that differences in patient characteristics in this small sample size may be a confounding factor affecting the results.

However, despite the small sample size, the patients were randomly assigned into the groups so as to minimize the effects of confounding factors. Moreover, the results of the subgroup analyses showed straightforward results; consistent superiority of group EC in terms of PFS (Suppl Fig. 1) and OS (Suppl Fig. 2) was shown. Therefore, we consider that discrepancies between the groups unlikely affected the present study findings indicating that the addition of capecitabine to maintenance therapy is beneficial in terms of PFS and OS.

Nevertheless, we understand that this is a phase 2 trial and the conclusions drawn are exploratory, and they need to be explored in a future phase 3 trial. We have revised the Limitations and Conclusions as follows:

3.5. Limitations and future research

“The present phase 2 trial was not designed to compare different combinations of endocrine agents and capecitabine. Also, it was not designed to investigate whether alternatives to capecitabine, such as CDK4/6 inhibitors, would be efficacious in combination with endocrine maintenance therapy, or whether bevacizumab should be added to maintenance therapy. It was aimed to determine which of the maintenance therapies (endocrine therapy alone or capecitabine plus endocrine therapy) should move on to phase 3, which will then compare the standard treatment (continuing initial chemotherapy until reaching PD) and chemotherapy followed by maintenance therapy.”

“As this was a phase 2 study, a limited number of patients were enrolled. However, patients were randomly assigned into the groups so as to minimize effects of confounding factors. Moreover, the results of the subgroup analyses showed straightforward results; consistent superiority of group EC in terms of PFS (Suppl Fig. 1) and OS (Suppl Fig. 2) was shown. Therefore, we consider that discrepancies between the groups unlikely affected the present study findings. A meta-analysis of data and a phase 3 study are needed to confirm the efficacy of the treatment strategy comprising induction, maintenance, and reinduction therapy in AMBC patients and establish optimal therapeutic strategies. Also, translational research analysis of serum samples collected at each evaluation point is still ongoing; the results will be presented in due course.”

Conclusions

“Our results suggest that the addition of capecitabine to maintenance endocrine therapy prolongs PFS and OS in patients with HR-positive, HER2-negative AMBC. To our knowledge, this is the first clinical trial to show the efficacy of endocrine therapy combined with metronomic capecitabine as maintenance therapy after induction chemotherapy in HR-positive, HER2-negative AMBC patients. Induction therapy followed by maintenance therapy could be a new therapeutic strategy for ABMC.”

Round 2

Reviewer 1 Report

The authors successfully addressed all raised issues. I only noticed several typos (e.g. page 2, lines 86-87 "has shown" should be replaced with "have shown", since the subject is plural; at page 11, line 377, a dot should precede "However" instead of a comma), therefore a careful revision is requested before final acceptance. 

Author Response

Thank you for your comment. We have corrected the above errors and checked throughout the manuscript.

Reviewer 2 Report

No additional comments

Author Response

Thank you for your comment and consideration of our manuscript.